# Comparison of standard and alternative methods for chest compressions in a single rescuer infant CPR: A prospective simulation study

**So Hyun Paek[1], Do Kyun Kim[2]\*, Jin Hee Lee[3], Young Ho Kwak[2,4]**

**1** CHA Bundang Medical Center, Department of Emergency Medicine, CHA University, Seongnam, Gyeonggi-do, Republic of Korea, **2** Department of Emergency Medicine, Seoul National University Hospital, Seoul, Republic of Korea, **3** Department of Emergency Medicine, Seoul National University Bundang Hospital, Gyeonggi-do, Republic of Korea, **4** College of Medicine, Department of Emergency Medicine, Seoul National University, Seoul, Republic of Korea

\* birdbeak@snuh.org

## Abstract

### Objective

The aims of this study were to develop a novel three-finger chest compression technique (pinch technique; PT) and an assistive device chest compression technique (plate-assisted technique; PAT) and compare these techniques with conventional techniques.

### Design

Prospective, crossover manikin study

### Setting

Pediatric emergency department at a tertiary care academic center

### Subjects

Fifty medical doctors and medical students

### Interventions

Using a manikin, fifty participants performed five different chest compression techniques—two 2-finger techniques (TFT1 and TFT2), two PTs (PT1 and PT2), and the PAT—for 2 minutes with 2 minutes of rest in a randomized sequence.

### Measurements and main results

The compression depth (CD), compression rate, recoil, and finger position were recorded. At the study conclusion, each participant completed a 5-point Likert scale-based questionnaire on fatigue, satisfaction and difficulty of performing each technique. The mean CDs were 32.9 mm (TFT1), 30.3 mm (TFT2), 37.3 mm (PT1), 35.0 mm (PT2) and 40.1 mm

**Data Availability Statement:** All relevant data are within the paper and its Supporting Information files.

**Funding:** The authors received no specific funding for this work.

**Competing interests:** The authors have declared that no competing interests exist.

(PAT) (p<0.001). TFT2 achieved the highest frequency of complete chest recoil, followed by PT1 and TFT1 (88.9%, 86.9%, and 81.4%, respectively, p = 0.003). The highest percentage of correct finger position was achieved by the PAT, followed by the PT1 and PT2 (93.4%, 83.1%, and 80.1%, respectively, p = 0.016). PAT use resulted in higher participant satisfaction, less fatigue, and less difficulty than the other four techniques.

## Conclusion

Our new chest compression methods using three fingers and assistive plates showed better CD results than the conventional 2-finger technique.

## Introduction

Pediatric cardiac arrest has a low rate of survival to hospital discharge compared to that of adults. The overall survival rate of out-of-hospital cardiac arrest in infants ranges from 6% to 27%, with poor neurological outcomes [1–3]. To improve the survival rates of pediatric cardiac arrest victims, current global guidelines, such as those of the European Resuscitation Council (ERC), American Heart Association (AHA), and Korean guideline for cardiopulmonary resuscitation (CPR), strongly recommend the delivery of high-quality chest compressions [4–6].

Both the ERC and AHA guidelines recommend using the 2-finger technique (TFT) in the center of the chest (the lower half of the sternum) for single-rescuer infant CPR, while for two or more rescuers, the 2-thumb encircling hands technique (TTHT) is recommended [4, 5]. Compared with the TFT, the TTHT was reported to consistently achieve higher blood and coronary perfusion pressure and greater compression depth (CD) in a variety of animal and manikin models [7–11]. However, most resuscitation guidelines recommend the TFT method for single-rescuer infant CPR because of the longer hands-off time and difficulty of postural change [11].

Several studies have investigated whether different methods deliver better chest compressions than TFT for single-rescuer infant CPR [12–15]. To overcome the insufficient CD of the TFT and alleviate discomfort in the user's fingers after prolonged compressions, we have developed two new compression methods: the pinch technique (PT) and the plate-assisted technique (PAT). The PT uses three fingers, which makes chest compressions easier. The PAT uses the thumb, index and middle fingers, along with a round plate. We performed a comparative analysis of two types of TFT, two types of PT and PAT during an infant CPR simulation.

## Materials and methods

The study was approved by the Institutional Review Board (approval IRB No: H-1606-048-770) of Seoul National University Hospital. Each participant received oral and written information about the study and the study protocol before providing written informed consent.

### Study design

This prospective manikin trial was conducted between October and November 2016 at a tertiary teaching hospital. A total of 50 individuals participated in the study, including 17 medical doctors (five interns and 12 emergency medicine residents) and 33 medical school students. All participants received basic life support (BLS) provider education. Chest compressions were performed using the five different techniques. TFT1 compression used the index and middle

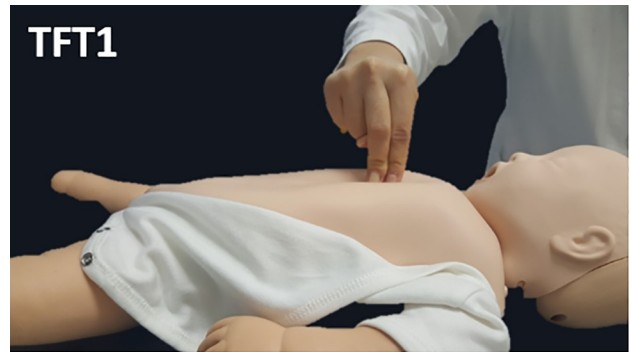
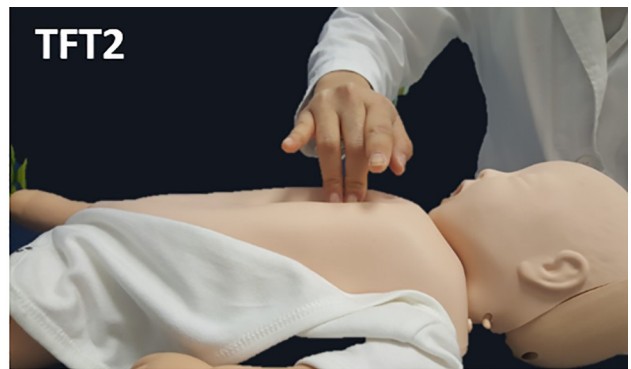
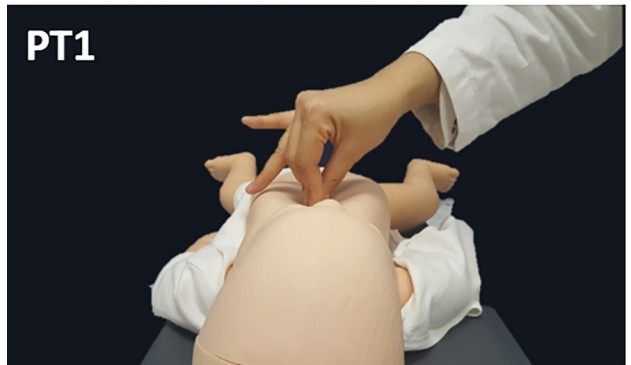
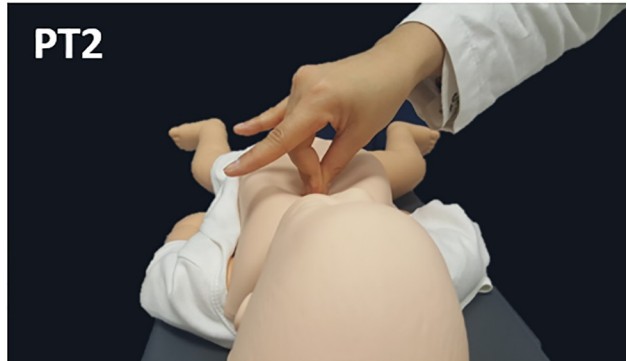
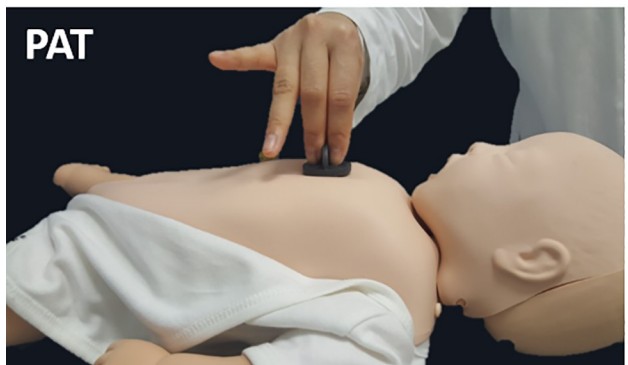

**Fig 1. Postures for the cardiac compressions.** TFT1: index-middle fingers; TFT2: middle-ring fingers; PT1: thumb-index-middle fingers; PT2: thumb-middle-ring fingers; PAT: thumb-index-middle fingers with plate.

fingers; TFT2 compression used the middle and ring fingers; PT1 compression used the thumb, index and middle fingers; PT2 compression used the thumb, middle and ring fingers; and PAT compression used the new device technique (Fig 1). The PAT device is made of rubber, is circular in shape and has a partition that divides the device into three areas for the thumb, index finger, and middle finger. A 1.0 cm in diameter circular piece is attached to the bottom of the device to simplify finding the correct compression point position (Fig 2). This device is made by the Department of Medical Engineering at Seoul National University as our request and is not patented and is not commercially available.

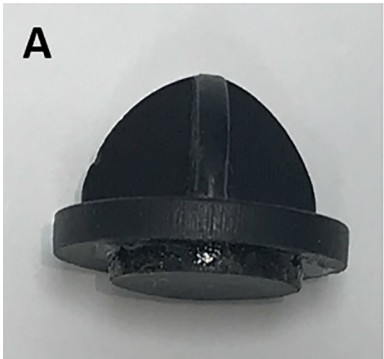 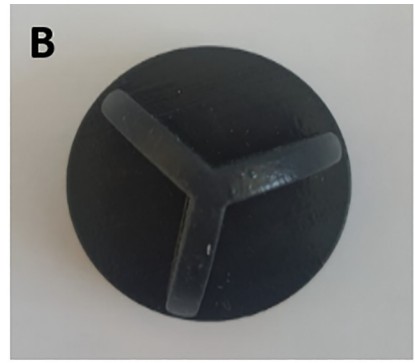 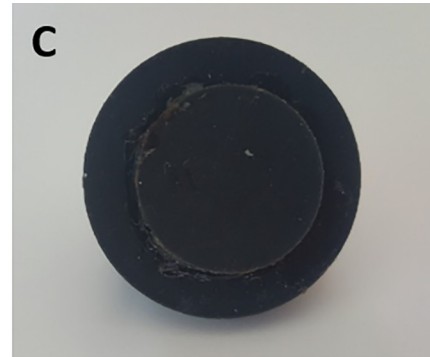

**Fig 2.** The new infant compression device (A: lateral view, B: viewed from above, C: viewed from below).

A 3-month-old infant manikin (Resusci® Baby QCPR® by Laerdal Medical) was used and was fitted with a SimPad PLUS with Skill Reporter (Laerdal Medical). The manikin was placed on a table that was 1.0 m high and reached the approximate waistline of the rescuers. Prior to conducting the study, all the participants received a 10-minute training session on compression rate, compression location and CD, according to current CPR guidelines using the standard TFT. Additionally, a brief introduction to the PT and PAT was given.

CPR was performed on the manikin for 2 minutes with each technique, followed by a 2-minute rest. The participants did not receive feedback regarding their performance during the study period. At the end of each technique, the participants completed the 5-point Likert scale-based questionnaire. The questionnaire recorded the fatigue, satisfaction and difficulty experienced by the rescuers.

### Measured outcomes

The primary outcome was the CD (mm) during CPR. The secondary outcomes were compression rate (compression/min), correct compression rate (%), correct CD (%), correct chest recoil (%), correct finger position (%), and the 5-point Likert scale-based questionnaire for satisfaction, fatigue and easiness.

### Data collection

The chest compression performance data were collected using the SimPad Skill Reporter. For each technique, the total number of chest compressions (compression/min), CD (mm), and the percentages of correct CD, adequate compression rate, compression with correct finger position, and complete recoil of the chest were recorded. The correct finger position was specified as just below the nipple line in the middle of the chest in a $1.0 \times 1.0$ cm square. We defined correct CD as $\geq 40$ mm, correct compression rate as 100–120/min, and complete recoil as the manikin's chest returning to its original position.

### Statistical analysis

The data were analyzed using the SPSS 22.0 statistical package (SPSS, Inc., Chicago, IL, USA). The values of continuous variables are expressed as the means ± standard deviation (SD). Linear mixed-effect models for repeated measures were used to compare the five chest compression methods. The fixed effects in the models were the compression methods, period, and interaction between the compression methods and period; the study participants were treated

as the random effect. A post hoc analysis was performed using the Bonferroni method. A Bonferroni corrected p-value <0.05 was considered statistically significant.

The sample size was calculated based on chest CD. A sample size of 46 rescuers provided an alpha of 0.05 and a power of 80%, which would allow detection of a between-subject difference of 0.6 SD, assuming a between-subject difference in depth of 4 mm. To account for 8% potential drop-out, we planned to enroll a total of 50 rescuers.

For the credible randomization of the serial sequential measurements of the different chest compression techniques, we used the Williams design with 5 x 5 crossovers. The Williams design is a special combination of crossover and Latin square designs [16].

## Results

The study participants included 50 medical doctors and medical school students (33 males; 66%), including 12 resident physicians (24%), 5 intern physicians (10%) and 33 medical students (66%). The mean participant age was 26.0 ± 3.2 years.

### Primary outcomes

The compression methods compared in this study are shown in Table 1. The effect of the interaction between compression method and period on the CD was not significant (p>0.05). The period was not significantly different among the compression methods (p = 0.28). The CD was significantly different among the compression methods (p<0.001). The mean CDs of each group were 32.9 mm (TFT1), 30.3 mm (TFT2), 37.3 mm (PT1), 35.0 mm (PT2) and 40.1 mm (PAT). The mean CD was best when PAT was used, while the TFT2 produced the lowest CD. All pairwise comparisons were significantly different for CD, except for TFT1 vs. PT2 and PT1 vs. PT2 (Fig 3A).

### Secondary outcomes

The interactions between protocol and period were not significant for any secondary outcomes (p>0.05). The PAT showed the best results for the compression indices (rate, correct rate, correct depth, and correct finger position), while PT1 exhibited the best results among the

**Table 1. Comparisons of chest compression quality of the five different techniques.**

| Compression Method Parameter | TFT1 | TFT2 | PT1 | PT2 | PAT | |
|---|---|---|---|---|---|---|
| | Mean ± SD | | | | | p-value |
| CPR Quality | | | | | | |
| Depth of compression (mm) | 32.9 ± 0.6 | 30.3 ± 0.7 | 37.3 ± 0.4 | 35.0 ± 0.6 | 40.1 ±0.3 | <0.001* |
| Rate of compression (compression/min) | 107.3 ± 9.3 | 103.3 ± 11.9 | 109.1 ± 7.8 | 107.8 ± 7.3 | 109.8 ± 7.8 | 0.004* |
| Correct rate (%) | 77.6 ± 34.5 | 78.5 ± 31.3 | 85.1 ± 27.0 | 82.4 ± 30.3 | 95.0 ± 8.3 | <0.001* |
| Correct depth (%) | 22.1 ± 31.3 | 13.2 ± 23.1 | 55.9 ± 24.1 | 29.7 ± 36.3 | 76.5 ±23.2 | <0.001* |
| Correct finger position (%) | 79.7 ± 34.3 | 75.8 ± 35.1 | 83.1 ± 28.3 | 80.1 ± 31.6 | 95.4 ±16.5 | 0.015* |
| Correct chest recoil (%) | 86.9 ± 25.4 | 92.8 ± 18.9 | 88.9 ± 18.6 | 81.4 ± 31.6 | 74.7 ± 28.9 | 0.003* |
| Questionnaires | | | | | | |
| Satisfaction | 2.6 ± 1.1 | 2.5 ± 0.9 | 3.2 ± 1.0 | 3.1 ± 0.9 | 3.8 ± 1.0 | <0.001* |
| Fatigue | 3.8 ± 1.0 | 3.9 ± 0.9 | 3.5 ± 0.8 | 3.6 ± 0.8 | 2.6 ± 0.9 | <0.001* |
| Difficulty | 2.5 ± 1.1 | 2.6 ± 0.9 | 3.1 ± 0.8 | 2.9 ± 0.9 | 4.0 ± 0.9 | <0.001* |

TFT: Two Finger Technique, PT: Pinch Technique, PAT: Plate-assisted Technique, CPR: Cardiopulmonary Resuscitation

*A statistically significant by linear mixed-effect mode

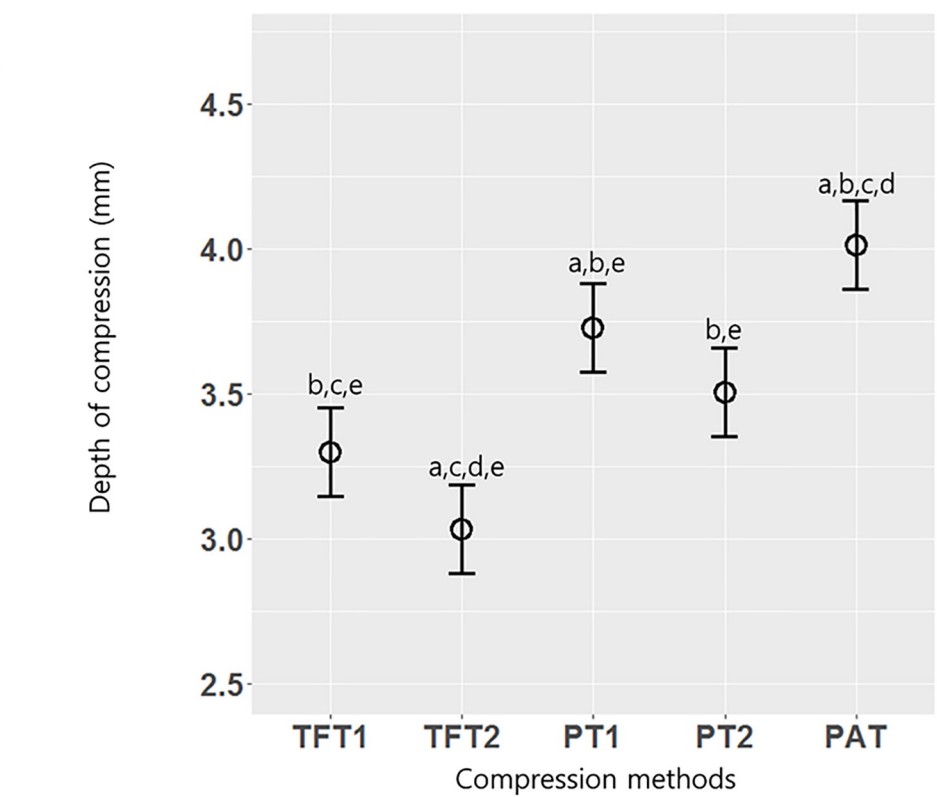

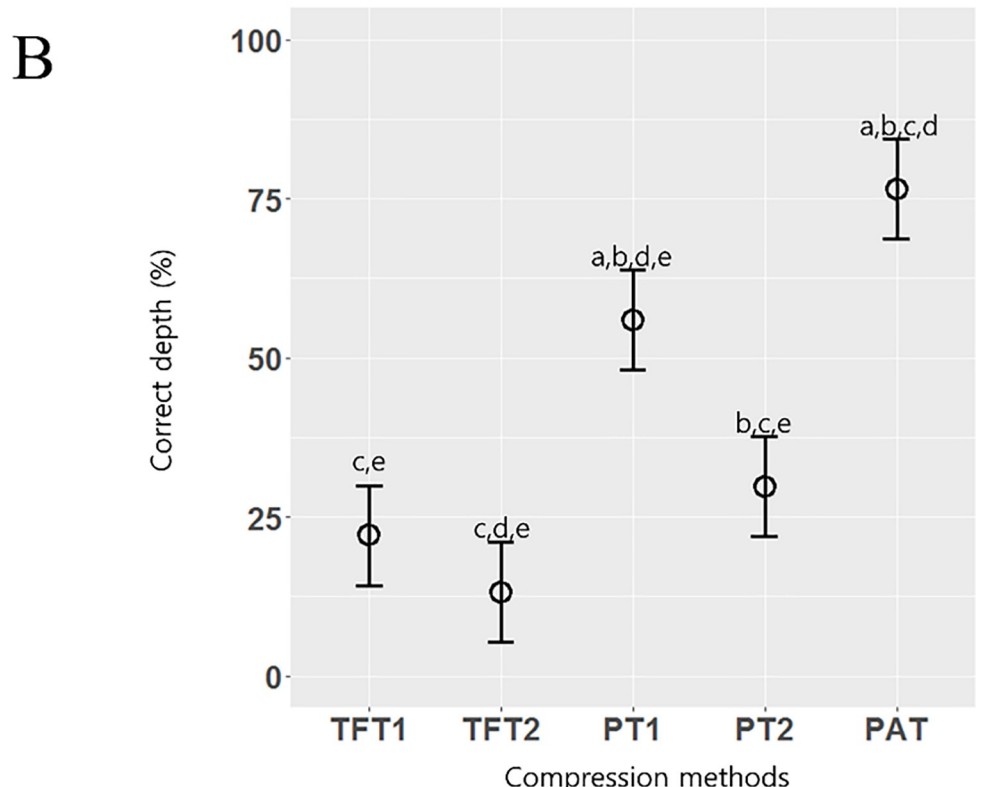

**Fig 3. Linear mixed-effect model.** The post hoc analysis was performed using a Bonferroni method. A Bonferroni corrected p-value <0.05 was considered statistically significant. A. Depth of compression (mm), B. Correct depth (%). a: TFT1, b: TFT2, c: PT1, d: PT2, e: PAT.

techniques that did not use a device (Table 1). Only the PAT achieved a mean target depth of ≥40 mm, which was significantly greater than the depths achieved with the other techniques (p<0.001, Fig 3B). All the techniques achieved a target compression rate of 100–120/min (p = 0.004). The highest percentage of chest recoil was recorded for TFT2 (92.8%), followed by PT1 (88.9%); the lowest percentage was achieved by the PAT (74.7%), and this difference was statistically significant (p = 0.003). The highest correct finger position percentage was recorded using the PAT (95.4%), followed by PT1 (83.1%), and the lowest percentage was achieved with TFT2 (75.8%) (Table 1).

The participants completed a questionnaire after each technique was performed. The questionnaire consisted of items regarding satisfaction with the cardiac compression, the degree of difficulty and the degree of fatigue; these were evaluated with a 5-point Likert scale (Table 1). The PAT had the highest satisfaction score, lowest difficulty of compression score and lowest fatigue score. TFT2 had the highest fatigue score. All items exhibited significant differences (Table 1). A post hoc analysis showed that the PAT obtained better scores than the other compression techniques for satisfaction, fatigue and easiness (Figures E, F, and G in S1 File).

## Discussion

In this study, we developed and evaluated a new compression technique (PT) and a new chest compression device (PAT). To the best of our knowledge, this is the first study to examine the use of a device for infant chest compressions performed by a single rescuer. When the five chest compression techniques were compared, the PAT exhibited the best results, followed by the PT1. According to our results, the PAT can be recommended for optimal chest compression. If the PAT is unavailable, the PT1—a technique using the index and middle fingers and the thumb—is recommended as a single-rescuer infant compression technique rather than conventional TFT.

For single-rescuer infant CPR, the TFT is currently recommended in global and regional CPR guidelines [4–6]. However, because the TFT method is less effective than the two-thumb method for chest compressions, various studies have examined the use of other chest compression methods to replace the two-finger method [12–15]. In this study, we proposed a PT that involved adding the thumb to the conventional TFT. The fatigue or discomfort that can occur when only two fingers are used seems to be relieved by adding the thumb. Furthermore, the development and application of a plate that can be grasped with three fingers, including the thumb, resulted in better chest compression indices and greater satisfaction than the conventional TFT and the newly developed PT. However, the recoil results of the chest compression index for the PAT were significantly lower than those of the PT and TFT.

As with adult BLS, it is important to maintain adequate chest CD when providing pediatric life support [4, 5, 17]. However, several manikin studies have shown that the TFT results in suboptimal CD, and some have shown that TFT only achieves a depth of approximately 25 mm [12, 13]. To address this problem, several new infant chest compression techniques and tools have been developed and evaluated. A study by Fakhraddin et al. reported that a new compression technique using the thumb and index finger improved the chest compression index results [12]. The thumb is more powerful than the other fingers; thus, use of the thumb is optimal for improving CD. The new technique has demonstrated good results that are similar to those of our study, particularly for achieving adequate CD and avoiding fatigue. The CD

of this new compression technique was similar to that of TTHT (33.3 mm vs. 33.1 mm) [12]. As with the Fakhraddin et al. [12] study, the PT proposed in our study was beneficial, as the results showed that the finger-pressing method that included the thumb without using an instrument was useful. In this study, the TFT was performed using only the index and middle fingers or the middle and ring fingers of the participant's right hand. When the chest pressure index was compared between the two methods used for the TFT, as in the previous study, the mean CD of the index and middle fingers technique was greater; however, the difference was not statistically significant (32.9 mm vs. 30.3 mm, p = 0.05). This result is similar to those of a previous study in which the right-hand index and middle fingers method showed the best results among several TFT methods [18].

A recent study of another new compression technique—the knocking-fingers technique—found high-quality chest compression and ventilation with minimal interruption of chest compressions [15], suggesting that it is an effective alternative chest compression technique for infant CPR. In a computed tomography-based study of a new compression method, the results suggested that the greater the compression area, the higher the risk of damage to the external organs [19]. The knocking-fingers technique had the smallest area of chest compression, with a median vertical length of 12 mm and a median horizontal length of 30 mm [15]. Although we used a related device, the compression area of the instrument used in our study was smaller than that of the knocking-fingers method, which could further reduce the risk of organ damage.

Both the traditional TFT and a new PT exhibited suboptimal CD, whereas the PAT showed a depth that was sufficient to recommend it. The technique with the highest percentage of correct finger positioning was the PAT, followed by the PT1; the TFT2 had the lowest percentage of correct finger position. The PAT showed the best results for depth, rate, correct compression rate, correct CD, and correct finger position. The participants felt at ease and were satisfied with the new device. The strengths of the PAT are as follows: 1) the compression position is limited to only the 1-cm diameter of the device, 2) it is easy to place the fingers and maintain the finger position, and 3) it is not easily displaced on the skin. Despite these advantages, the PAT showed the poorest result in terms of compression relaxation among the tested techniques. It may be difficult to determine sufficient relaxation because the rescuer makes contact with the skin indirectly, through the plate. When using the PAT, it is important to emphasize that the rescuer should not lean on the plate when the chest is recoiling.

If the chest recoil index results for the PAT could be improved, we would recommend the PAT as the best method for single-rescuer infant CPR. However, the effect of this method has been confirmed only in one well-designed manikin study; therefore, verification in a clinical setting or animal model is needed.

## Limitations

Our study had some limitations. First, the chest stiffness and resistance and size of manikin models may not accurately represent these characteristics in humans, and the experimental conditions may not accurately represent real-life situations. In addition, it is possible that the results would be different under more stressful conditions. Second, we did not measure the pressure quality, as some simulation studies have, to assess compression quality [7, 10]. However, the usefulness of the chest compression indices recorded during infant manikin tests has been confirmed in several studies, and the results of our study are considered reliable under these circumstances [8–11, 15, 18]. Third, rescue ventilations were not included in our study protocol. Because the purpose of this study was to compare five different chest compression techniques and the chest compression indices of single-rescuer infant CPR, the study was

conducted without using rescue ventilation. Fourth, the participants varied and included residents, interns, and medical students; however, no significant difference was found when comparing the quality indices of the three groups (data not shown). The participants were likely to have similar characteristics given that they did not have experience with infant CPR and were all within 2 years of obtaining their BLS certification. Finally, for future studies to evaluate the effectiveness of the techniques, the participants should include experienced doctors and healthcare practitioners since the current study only included medical students and residents.

## Conclusions

Our new PT and PAT showed better results for CD than the conventional TFT. The PAT resulted in less fatigue for the rescuers and superior CD compared to the other techniques. However, the results were from only manikin models, so external validation of the above findings is needed.

## Supporting information

**S1 File.** Figure A. Comparison of rate of compression (/mm) between five methods.
Figure B. Comparison of correct rate (%) between five methods.
Figure C. Comparison of correct finger position (%) between five methods.
Figure D. Comparison of correct chest recoil (%) between five methods.
Figure E. Comparison of satisfaction score between five methods.
Figure F. Comparison of fatigue score between five methods.
Figure G. Comparison of easiness score between five methods.
(DOCX)

## Author Contributions

**Conceptualization:** Do Kyun Kim.

**Formal analysis:** So Hyun Paek.

**Investigation:** So Hyun Paek.

**Supervision:** Do Kyun Kim, Jin Hee Lee, Young Ho Kwak.

**Validation:** Jin Hee Lee, Young Ho Kwak.

**Writing – original draft:** So Hyun Paek.

**Writing – review & editing:** Do Kyun Kim.

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
