## [Decision Letter · Decision Letter 0]

3 Sep 2019

PONE-D-19-19476

Comparison of standard and alternative methods for chest compressions in a single rescuer infant CPR: a prospective simulation study

PLOS ONE

Dear Dr Kim,

Thank you for submitting your manuscript to PLOS ONE. After careful consideration, we feel that it has merit but does not fully meet PLOS ONE’s publication criteria as it currently stands. Therefore, we invite you to submit a revised version of the manuscript that addresses the points raised during the review process.

We would appreciate receiving your revised manuscript by Oct 18 2019 11:59PM. To enhance the reproducibility of your results, we recommend that if applicable you deposit your laboratory protocols in protocols.io, where a protocol can be assigned its own identifier (DOI) such that it can be cited independently in the future. For instructions see: http://journals.plos.org/plosone/s/submission-guidelines#loc-laboratory-protocols

We look forward to receiving your revised manuscript.

Kind regards,

Lars-Peter Kamolz, M.D., Ph.D., M.Sc.

Academic Editor

PLOS ONE

Reviewers' comments:

Reviewer's Responses to Questions

**Comments to the Author**

1. Is the manuscript technically sound, and do the data support the conclusions?

Reviewer #1: Yes

Reviewer #2: Yes

2. Has the statistical analysis been performed appropriately and rigorously? 

Reviewer #1: Yes

Reviewer #2: Yes

3. Have the authors made all data underlying the findings in their manuscript fully available?

Reviewer #1: Yes

Reviewer #2: Yes

4. Is the manuscript presented in an intelligible fashion and written in standard English?

Reviewer #1: Yes

Reviewer #2: Yes

5. Review Comments to the Author

Reviewer #1: After reviewing the PONE-D-19-19476 manuscript entitled “Comparison of standard and alternative methods for chest compressions in a single rescuer infant CPR: a prospective simulation study” some suggestions and clarifications were raised as follows:

The aim of the study was to assess a new technique/device against 4 different techniques on a simulation manikin. In general, the manuscript is well written with minor spelling or grammar mistakes.

After checking the literature, and to my knowledge, this is the only manuscript that evaluated the use of a devise to assist the rescuers in performing CPR. This is a vital area of research since improving CPR in pediatric would rescue many lives.

Please find below some of the suggestions that would improve the quality of the manuscript.

Major suggestions or questions:

1. Why did it take the investigators a long time after the simulation (Oct and Nov 2016) to prepare the manuscript and submit it for publications? Many papers were published during this period on new finger location and numbers since then. (1-5) one of the new techniques has many publications showing its superiority to TFT.

2. The authors used a random cross over method to test the different techniques. Also they mentioned that some of the techniques caused fatigue more than the others. Did the authors check the effect of starting with a harder technique on the performance of participants in the other techniques?

3. Did the authors check the difference in the performance of the participants depending on their levels (doctors, residents and students)?

4. How were the CD, the compression rate, finger position, and the recoil recorded and evaluated? Does the manikin record them?

5. Does the manikin record other parameters like simulated SBP, DBP, mean arterial pressure, and pulse pressure?

6. Did the author account for the primary/dominant hand of the participant? In the text they mentioned that all the participants used their right hand fingers for the compressions. How many participants were left handed? How was the performance of these participants in comparison with the right handed ones?

7. The authors should provide further description or details about the used device in the PAT technique.

a. Is it patented?

b. Is it commercially available? What company invented it?

c. Is there any other device in the market or experimental to compare this device to it?

8. Depending on the author responses to question 7, the conflict of interest of the group would be more clear to the reviewers and the journal.

Minor

1. Spelling mistakes (line 2 Paediatric and line 4, pediatric).

2. On line 145, remove one of “That” after TFT.

3. The description of the dimensions of the circular piece (line 39) must report either the diameter or the radius and not 2 dimensions since it is circle shape. Please modify.

4. Authors must improve the appearance of Table 1. It is hard to get the results as it is now.

5. The orientation of the manikin and the fingers position in figure 1 (PT1 and PT2) is not clear. I suggest a better orientation like the other images.

6. I suggest that the authors move some of the graphs presented in the supplementary sections to the main manuscript body to enrich the article.

7. The numbering of the figures in the supporting information file should start with 1, and the authors should assign a special character to reflect the location of the figure in the supplement file. The authors should check the guidelines of PLoS one for this issue.

1. Ladny J, Smereka J, Rodriguez-Nunez A, Leung S, Ruetzler K, Szarpak L. Is there any alternative to standard chest compression techniques in infants? A randomized manikin trial of the new “2-thumb-fist” option. 2018. e9386 p.

2. Smereka J, Bielski K, Ladny J, Ruetzler K, Szarpak L. Evaluation of a newly developed infant chest compression technique A randomized crossover manikin trial 2017. e5915 p.

3. Smereka J, Szarpak L, Ladny J, Rodriguez-Nunez A, Ruetzler K. A Novel Method of Newborn Chest Compression: A Randomized Crossover Simulation Study 2018. 159 p.

4. Smereka J, Szarpak L, Rodriguez-Nunez A, Ladny J, Leung S, Ruetzler K. A randomized comparison of three chest compression techniques and associated hemodynamic effect during infant CPR: A randomized manikin study 2017. 1420-5 p.

5. Smereka J, Szarpak L, Smereka A, Leung S, Ruetzler K. 2017. Evaluation of new two-thumb chest compression technique for infant CPR performed by novice physicians. A randomized, crossover, manikin trial 604-9 p.

Reviewer #2: Dear authors,

Thank you for the opportunity to review the manuscript "Comparison of standard and alternative methods for chest compressions in a single rescuer infant CPR: a prospective simulation study".

A very interesting study concerning an important medical field that I have read with great interest.

I wish you all the best for publication.

Kind regards

6. PLOS authors have the option to publish the peer review history of their article (what does this mean?). If published, this will include your full peer review and any attached files.

Reviewer #1: Yes: Moustafa Al Hariri

Reviewer #2: No

---

## [Author Response · Author response to Decision Letter 0]

2 Oct 2019

Response to Reviewers

We would like to thank the reviewer for the careful and thorough reading of this manuscript and for the thoughtful comments and constructive suggestions.

Comments to the Author

1. Is the manuscript technically sound, and do the data support the conclusions?

Reviewer #1: Yes

Reviewer #2: Yes 

2. Has the statistical analysis been performed appropriately and rigorously? 

Reviewer #1: Yes

Reviewer #2: Yes 

3. Have the authors made all data underlying the findings in their manuscript fully available?

Reviewer #1: Yes

Reviewer #2: Yes

4. Is the manuscript presented in an intelligible fashion and written in standard English?

Reviewer #1: Yes

Reviewer #2: Yes 

5. Review Comments to the Author

Reviewer #1: After reviewing the PONE-D-19-19476 manuscript entitled “Comparison of standard and alternative methods for chest compressions in a single rescuer infant CPR: a prospective simulation study” some suggestions and clarifications were raised as follows:

The aim of the study was to assess a new technique/device against 4 different techniques on a simulation manikin. In general, the manuscript is well written with minor spelling or grammar mistakes.

After checking the literature, and to my knowledge, this is the only manuscript that evaluated the use of a devise to assist the rescuers in performing CPR. This is a vital area of research since improving CPR in pediatric would rescue many lives.

Please find below some of the suggestions that would improve the quality of the manuscript.

Major suggestions or questions:

1. Why did it take the investigators a long time after the simulation (Oct and Nov 2016) to prepare the manuscript and submit it for publications? Many papers were published during this period on new finger location and numbers since then. (1-5) one of the new techniques has many publications showing its superiority to TFT.

- Thank you for your comment. I was able to finish the prospective study in a timely manner but due to personal health reasons, I was not able to finish writing the paper. Now that I have almost full recovered, I had the time to finally write it. I am aware that there were other papers published on new devices with a mannequin study, so I was deeply disappointed in my late publication. Thus, I would really love to have it published in your journal.

.

2. The authors used a random cross over method to test the different techniques. Also, they mentioned that some of the techniques caused fatigue more than the others. Did the authors check the effect of starting with a harder technique on the performance of participants in the other techniques?

Response: Thank you for your comment. We did not check the effect of starting with a harder technique because we purposely used a random order for the different chest compression techniques for each participant. Thus, we felt that starting with a harder technique was not necessary. Also, CPR was performed on the manikin for 2 minutes with each technique, followed by a 2-minute rest. This 2-minute rest time gave the participants ample time to recover from any fatigue. 

3. Did the authors check the difference in the performance of the participants depending on their levels (doctors, residents and students)?

Response: Thank you for your comment. We did check the performance based upon their levels, but the data showed that it was not statistically significant. This point is already described in ‘Limitation’ session.

4. How were the CD, the compression rate, finger position, and the recoil recorded and evaluated? Does the manikin record them?

- Thank you for your comment. In the Study Design section of the Materials and Methods it is explained that a 3-month-old infant manikin (Resusci® Baby QCPR® by Laerdal Medical) was used and was fitted with a SimPad PLUS with Skill Reporter (Laerdal Medical). The chest compression performance data were collected using the SimPad Skill Reporter. For each technique, the total number of chest compressions (compression/min), CD (mm), and the percentages of correct CD, adequate compression rate, compression with correct finger position, and complete recoil of the chest were recorded. The analyzed comparison data among each different method groups are presented in Table 1.

5. Does the manikin record other parameters like simulated SBP, DBP, mean arterial pressure, and pulse pressure?

- Thank you for your comment. The manikin cannot record other parameters such as simulated SBP, DBP, mean arterial pressure, and pulse pressure. In the future, after the publication of this article, we plan to use an infant animal model to test the effectiveness of the new device and record other parameters.

6. Did the author account for the primary/dominant hand of the participant? In the text they mentioned that all the participants used their right hand fingers for the compressions. How many participants were left handed? How was the performance of these participants in comparison with the right handed ones?

- Thank you for your comment. Two of the participants were left-handed and were given the opportunity to use their dominant hand but both of them individually chose to use their right hand for chest compressions. 

7. The authors should provide further description or details about the used device in the PAT technique.

a. Is it patented? 

- The used device is not patented.

b. Is it commercially available? What company invented it? 

- It is not commercially available. Three samples were made by the Department of Medical Engineering at Seoul National University as our request. 

c. Is there any other device in the market or experimental to compare this device to it?

- There is no other device in the market or experimental to compare with this device.

8. Depending on the author responses to question 7, the conflict of interest of the group would be more clear to the reviewers and the journal.

- Thank you for your comment. We have added the above to the COI. 

Minor

1. Spelling mistakes (line 2 Paediatric and line 4, pediatric). 

- We made the spelling uniform.

2. On line 145, remove one of “That” after TFT. 

- The word was removed.

3. The description of the dimensions of the circular piece (line 39) must report either the diameter or the radius and not 2 dimensions since it is circle shape. Please modify. 

- We modified it to “a 1.0 cm in diameter…”

4. Authors must improve the appearance of Table 1. It is hard to get the results as it is now. 

- The table has been modified.

5. The orientation of the manikin and the fingers position in figure 1 (PT1 and PT2) is not clear. I suggest a better orientation like the other images. 

- Thank you for your comment. However, if we take pictures from another orientation, the position of thumb is not clearly visible so it is difficult to distinguish finger position.

6. I suggest that the authors move some of the graphs presented in the supplementary sections to the main manuscript body to enrich the article. 

- Your suggestion has been noted. Among supplementary figures, two figures was moved as Fig 3.

7. The numbering of the figures in the supporting information file should start with 1, and the authors should assign a special character to reflect the location of the figure in the supplement file. The authors should check the guidelines of PLoS one for this issue. 

- The numbering of the figures was adjusted according to the guidelines of PLoS one.

1. Ladny J, Smereka J, Rodriguez-Nunez A, Leung S, Ruetzler K, Szarpak L. Is there any alternative to standard chest compression techniques in infants? A randomized manikin trial of the new “2-thumb-fist” option. 2018. e9386 p.

2. Smereka J, Bielski K, Ladny J, Ruetzler K, Szarpak L. Evaluation of a newly developed infant chest compression technique A randomized crossover manikin trial 2017. e5915 p.

3. Smereka J, Szarpak L, Ladny J, Rodriguez-Nunez A, Ruetzler K. A Novel Method of Newborn Chest Compression: A Randomized Crossover Simulation Study 2018. 159 p.

4. Smereka J, Szarpak L, Rodriguez-Nunez A, Ladny J, Leung S, Ruetzler K. A randomized comparison of three chest compression techniques and associated hemodynamic effect during infant CPR: A randomized manikin study 2017. 1420-5 p.

5. Smereka J, Szarpak L, Smereka A, Leung S, Ruetzler K. 2017. Evaluation of new two-thumb chest compression technique for infant CPR performed by novice physicians. A randomized, crossover, manikin trial 604-9 p.

Reviewer #2: Dear authors,

Thank you for the opportunity to review the manuscript "Comparison of standard and alternative methods for chest compressions in a single rescuer infant CPR: a prospective simulation study".

A very interesting study concerning an important medical field that I have read with great interest.

I wish you all the best for publication.

Kind regards

6. PLOS authors have the option to publish the peer review history of their article (what does this mean?). If published, this will include your full peer review and any attached files.

Do you want your identity to be public for this peer review? For information about this choice, including consent withdrawal, please see our Privacy Policy.

Reviewer #1: Yes: Moustafa Al Hariri

Reviewer #2: No

---

## [Decision Letter · Decision Letter 1]

31 Oct 2019

PONE-D-19-19476R1

Comparison of standard and alternative methods for chest compressions in a single rescuer infant CPR: a prospective simulation study

PLOS ONE

Dear Dr Kim,

Thank you for submitting your manuscript to PLOS ONE. After careful consideration, we feel that it has merit but does not fully meet PLOS ONE’s publication criteria as it currently stands. Therefore, we invite you to submit a revised version of the manuscript that addresses the points raised during the review process.

We would appreciate receiving your revised manuscript by Dec 15 2019 11:59PM. To enhance the reproducibility of your results, we recommend that if applicable you deposit your laboratory protocols in protocols.io, where a protocol can be assigned its own identifier (DOI) such that it can be cited independently in the future. For instructions see: http://journals.plos.org/plosone/s/submission-guidelines#loc-laboratory-protocols

We look forward to receiving your revised manuscript.

Kind regards,

Lars-Peter Kamolz, M.D., Ph.D., M.Sc.

Academic Editor

PLOS ONE

Reviewers' comments:

Reviewer's Responses to Questions

**Comments to the Author**

1. If the authors have adequately addressed your comments raised in a previous round of review and you feel that this manuscript is now acceptable for publication, you may indicate that here to bypass the “Comments to the Author” section, enter your conflict of interest statement in the “Confidential to Editor” section, and submit your "Accept" recommendation.

Reviewer #2: (No Response)

Reviewer #3: (No Response)

2. Is the manuscript technically sound, and do the data support the conclusions?

Reviewer #2: (No Response)

Reviewer #3: Yes

3. Has the statistical analysis been performed appropriately and rigorously? 

Reviewer #2: (No Response)

Reviewer #3: Yes

4. Have the authors made all data underlying the findings in their manuscript fully available?

Reviewer #2: (No Response)

Reviewer #3: Yes

5. Is the manuscript presented in an intelligible fashion and written in standard English?

Reviewer #2: (No Response)

Reviewer #3: Yes

6. Review Comments to the Author

Reviewer #2: (No Response)

Reviewer #3: Dear authors,

I got to review your manuscript "Comparison of standard and alternative methods for chest compressions in a single rescuer infant CPR: a prospective simulation study", in which you show the superiority of a newly developed technique over the currently recommended ones. While in general, I think your manuscript is of great value and has gained a lot of value by the first revision, I would like to add a few minor points:

1.) Consider adding another figure about the currently recommended techniques (lines 8ff.) for the purpose of presentation.

2.) I would appreciate a little more context about your motivation, on WHY you performed this research. You mention, that you wanted to "overcome the insufficient CD of the TFT and alleviate discomfort", however, please elaborate a little bit more, especially considering the fact THAT and why other techniques seem to be insufficient.

3.)In line 51, you should mention that the 5-point Likert scale questionnaire was performed after EVERY one of the techniques. As it is written now, it appears that it was performed just ones at the very end (In line 118 you mention it correctly).

4.) In the measured outcomes section you write about all the outcomes, however, it remains unclear how "correct" is defined. in Data Collection you write about the correct finger position, but how correct (or adequate) compression rate, correct chest recoil and correct CD are defined, remains unclear. Please clarify.

5.) I would be very interested, whether the sequence yielded different results for the different techniques (e.g. has the TFT1 technique yielded better results when performed first...?) I am wondering, because I can imagine, that when performing five different techniques, the motivation reduces over time. Furthermore, it would be also very interesting how the "correctness" of each technique changed over the duration of the performance - can any statement concerning the recommended length before a change of rescucitator be given? How significant is the fatigueness during the performance?

6.) Please add a sentence to the limitations section that this experiment should also be repeated with a group of experienced doctors/health care practitioners (like paediatricians or emergency doctors) to evaluate the effectiveness in a group of other relevance.

Thank you.

7. PLOS authors have the option to publish the peer review history of their article (what does this mean?). If published, this will include your full peer review and any attached files.

Reviewer #2: No

Reviewer #3: No

---

## [Author Response · Author response to Decision Letter 1]

18 Nov 2019

Response to Reviewers

Comments to the Author

1. If the authors have adequately addressed your comments raised in a previous round of review and you feel that this manuscript is now acceptable for publication, you may indicate that here to bypass the “Comments to the Author” section, enter your conflict of interest statement in the “Confidential to Editor” section, and submit your "Accept" recommendation.

Reviewer #2: (No Response)

Reviewer #3: (No Response)

2. Is the manuscript technically sound, and do the data support the conclusions?

Reviewer #2: (No Response)

Reviewer #3: Yes

3. Has the statistical analysis been performed appropriately and rigorously? 

Reviewer #2: (No Response)

Reviewer #3: Yes

4. Have the authors made all data underlying the findings in their manuscript fully available?

Reviewer #2: (No Response)

Reviewer #3: Yes

5. Is the manuscript presented in an intelligible fashion and written in standard English?

Reviewer #2: (No Response)

Reviewer #3: Yes

6. Review Comments to the Author

Reviewer #2: (No Response)

Reviewer #3: Dear authors,

I got to review your manuscript "Comparison of standard and alternative methods for chest compressions in a single rescuer infant CPR: a prospective simulation study", in which you show the superiority of a newly developed technique over the currently recommended ones. While in general, I think your manuscript is of great value and has gained a lot of value by the first revision, I would like to add a few minor points:

1.) Consider adding another figure about the currently recommended techniques (lines 8ff.) for the purpose of presentation.

Thank you for your comment. The currently recommended one rescuer compression technique by the ERC and AHA guidelines is the 2-finger technique (TFT). TFT is shown as the first image in Figure 1 (TFT1). 

2.) I would appreciate a little more context about your motivation, on WHY you performed this research. You mention, that you wanted to "overcome the insufficient CD of the TFT and alleviate discomfort", however, please elaborate a little bit more, especially considering the fact THAT and why other techniques seem to be insufficient.

Thank you for your comment. The manuscript was adjusted as following: 

We undertook this study due to TFT having insufficient CD and to alleviate discomfort in the user’s fingers after prolonged compressions. TTHT was reported to consistently achieve higher blood and coronary perfusion pressure and greater compression depth (CD) in a variety of animal and manikin models (7-11). However, most resuscitation guidelines recommend the TFT method for single-rescuer infant CPR because of the longer hands-off time and difficulty of postural change.

3.) In line 51, you should mention that the 5-point Likert scale questionnaire was performed after EVERY one of the techniques. As it is written now, it appears that it was performed just ones at the very end (In line 118 you mention it correctly).

Thank you for your comment. Line 51 was changed to the following:

At the end of each technique, the participants completed the 5-point Likert scale-based questionnaire.

4.) In the measured outcomes section you write about all the outcomes, however, it remains unclear how "correct" is defined. in Data Collection you write about the correct finger position, but how correct (or adequate) compression rate, correct chest recoil and correct CD are defined, remains unclear. Please clarify.

Thank you for your comment. 

Lines 65 – 68 were adjusted to include the word “correct” as follows:

The correct finger position was specified as just below the nipple line in the middle of the chest in a 1.0 × 1.0 cm square. We defined correct CD as ≥ 40 mm, correct compression rate as 100–120/min, and complete recoil as the manikin’s chest returning to its original position.

5.) I would be very interested, whether the sequence yielded different results for the different techniques (e.g. has the TFT1 technique yielded better results when performed first...?) I am wondering, because I can imagine, that when performing five different techniques, the motivation reduces over time. Furthermore, it would be also very interesting how the "correctness" of each technique changed over the duration of the performance - can any statement concerning the recommended length before a change of rescucitator be given? How significant is the fatigueness during the performance?

Thank you for your comment. We did not check the effect of starting with a harder technique because we purposely used a random order for the different chest compression techniques for each participant. Thus, we felt that starting with a harder technique was not necessary. Also, chest compressions were performed on the manikin for 2 minutes with each technique, followed by a 2-minute rest. This 2-minute rest time gave the participants ample time to recover from any fatigue. A method of chest compression for 2-minute is widely recommended before changing the resuscitator. 

6.) Please add a sentence to the limitations section that this experiment should also be repeated with a group of experienced doctors/health care practitioners (like paediatricians or emergency doctors) to evaluate the effectiveness in a group of other relevance.

Thank you for your comment. This sentence was added to the Limitations section:

Finally, for future studies to evaluate the effectiveness of the techniques, the participants should include experienced doctors and healthcare practitioners since the current study only included medical students and residents. 

Thank you.

7. PLOS authors have the option to publish the peer review history of their article (what does this mean?). If published, this will include your full peer review and any attached files.

Do you want your identity to be public for this peer review? For information about this choice, including consent withdrawal, please see our Privacy Policy.

Reviewer #2: No

Reviewer #3: No

---

## [Decision Letter · Decision Letter 2]

4 Dec 2019

Comparison of standard and alternative methods for chest compressions in a single rescuer infant CPR: a prospective simulation study

PONE-D-19-19476R2

Dear Dr. Kim,

We are pleased to inform you that your manuscript has been judged scientifically suitable for publication and will be formally accepted for publication once it complies with all outstanding technical requirements.

With kind regards,

Lars-Peter Kamolz, M.D., Ph.D., M.Sc.

Academic Editor

PLOS ONE

Additional Editor Comments (optional):

Reviewers' comments:

Reviewer's Responses to Questions

**Comments to the Author**

1. If the authors have adequately addressed your comments raised in a previous round of review and you feel that this manuscript is now acceptable for publication, you may indicate that here to bypass the “Comments to the Author” section, enter your conflict of interest statement in the “Confidential to Editor” section, and submit your "Accept" recommendation.

Reviewer #2: (No Response)

Reviewer #3: All comments have been addressed

2. Is the manuscript technically sound, and do the data support the conclusions?

Reviewer #2: (No Response)

Reviewer #3: (No Response)

3. Has the statistical analysis been performed appropriately and rigorously? 

Reviewer #2: (No Response)

Reviewer #3: (No Response)

4. Have the authors made all data underlying the findings in their manuscript fully available?

Reviewer #2: (No Response)

Reviewer #3: (No Response)

5. Is the manuscript presented in an intelligible fashion and written in standard English?

Reviewer #2: (No Response)

Reviewer #3: (No Response)

6. Review Comments to the Author

Reviewer #2: (No Response)

Reviewer #3: (No Response)

7. PLOS authors have the option to publish the peer review history of their article (what does this mean?). If published, this will include your full peer review and any attached files.

Reviewer #2: No

Reviewer #3: No

---

## [Editor Report · Acceptance letter]

11 Dec 2019

PONE-D-19-19476R2 

Comparison of standard and alternative methods for chest compressions in a single rescuer infant CPR: a prospective simulation study 

Dear Dr. Kim:

I am pleased to inform you that your manuscript has been deemed suitable for publication in PLOS ONE. Congratulations! Your manuscript is now with our production department. 

With kind regards,

on behalf of

Dr. Lars-Peter Kamolz 

Academic Editor

PLOS ONE